# The Effect of Batter Characteristics on Protein-Aided Control of Fat Absorption in Deep-Fried Breaded Fish Nuggets

**DOI:** 10.3390/foods11020147

**Published:** 2022-01-06

**Authors:** Lulu Cui, Jiwang Chen, Yuhuan Wang, Youling L. Xiong

**Affiliations:** 1College of Food Science and Engineering, Wuhan Polytechnic University, Wuhan 430023, China; cuilulu190203@163.com (L.C.); yuhuanwang08@126.com (Y.W.); 2Key Laboratory for Deep Processing of Major Grain and Oil, Wuhan Polytechnic University, Wuhan 430023, China; 3Department of Animal & Food Sciences, University of Kentucky, Lexington, KY 40546, USA

**Keywords:** batter-breaded fish nuggets, protein, rheological behavior, thermal properties, fat absorption

## Abstract

Soy protein (SP), egg white protein (EP), and whey protein (WP) at 6% *w*/*w* were individually incorporated into the batter of a wheat starch (WS) and wheat gluten (WG) blend (11:1 *w*/*w* ratio). Moisture adsorption isotherms of WS and proteins and the viscosity, rheological behavior, and calorimetric properties of the batters were measured. Batter-breaded fish nuggets (BBFNs) were fried at 170 °C for 40 s followed by 190 °C for 30 s, and pick-up of BBFNs, thermogravimetric properties of crust, and fat absorption were determined. The moisture absorption capacity was the greatest for WS, followed by WG, SP, EP, and WP. The addition of SP significantly increased the viscosity and shear moduli (G″, G′) of batter and pick-up of BBFNs, while EP and WP exerted the opposite effect (*p* < 0.05). SP, EP, and WP raised WS gelatinization and protein denaturation temperatures and crust thermogravimetry temperature, but decreased enthalpy change (ΔH) and oily characteristics of fried BBFNs. These results indicate that hydrophilicity and hydration activity of the added proteins and their interactions with batter matrix starch and gluten reinforced the batter and the thermal stability of crust, thereby inhibiting fat absorption of the BBFNs during deep-fat frying.

## 1. Introduction

Fried batter-breaded foods are widely consumed food commodities. Aside from the tenderness and juiciness of the core (e.g., poultry, aquatic, and vegetable), the crispness of the crust and its golden yellow color, which depend on the composition of the breading batter (flour or starch, water, seasonings, and protein), are highly regarded by the consumers [1,2,3,4]. However, the crust formed during deep-fat frying contains significant amounts of fats, reaching one-third of fried batter-breaded foods by weight and in some cases up to 50% [5,6,7,8]. Long-term consumption of high-fat fried batter-breaded foods has the potential of contributing to cardiovascular and cerebrovascular diseases, high blood pressure, and dyslipidemia [5,7,9]. Therefore, strategies for creating low-fat fried batter-breaded foods that retain their desirable organoleptic properties are of importance.

Functional ingredients (proteins, non-protein hydrocolloids, and insoluble dietary fibers) can be added to the batter to inhibit oil penetration and moisture evaporation during deep-fat frying, resulting in decreased fat absorption [1,9,10,11,12]. Ingredient-based fat reduction strategies have focused on the proteins due to their film-forming and thermal gelation properties [10]. Moreover, the addition of proteins promotes emulsification and Maillard reactions of batter to improve quality attributes of fried batter-breaded foods such as color and crispiness [13,14]. Furthermore, a batter with high protein content renders a more nutritious coating [13].

Individual protein or protein mixtures from both animal and plant sources have been incorporated into batters to reduce fat absorption of fried batter-breaded muscle-based foods. Dogan et al. [15] reported that fat absorption of fried batter-breaded chicken nuggets was inhibited through adding egg albumen, soy protein (SP), and whey protein (WP) in the batter. Furthermore, the added egg albumen and WP significantly increased the moisture retention, facilitated color formation, and provided the crunchiest crust. Chen et al. [6] also found that the crust crispness of fried batter-breaded fish nuggets (BBFNs) containing 1% wheat protein or 1% SP was higher than the control (without protein added to the batter). With a batter of wheat starch (WS)/wheat protein blend (11:1 *w*/*w*), the fried BBFNs had the highest moisture content and the lowest fat content among five WS-to-wheat protein ratios [16].

In our previous study, adding SP, egg white protein (EP), and WP to the batter proved effective in reducing fat absorption of fried BBFNs. However, it is not clear how these individually added proteins affected batter characteristics and reduced fat absorption of fried BBFNs. Therefore, the objective of this work was to investigate the effect of adding different proteins to a starch-based batter on the rheological behavior and calorimetric properties of batters, pick-up of BBFNs, thermogravimetric properties of the crust, and fat absorption. These product characteristics were related to the moisture adsorption isotherms of starch and protein components used in the batter formulations.

## 2. Materials and Methods

### 2.1. Materials

Frozen silver carp surimi with 3% of sodium tripolyphosphate was obtained from Honghu Xinhongye Food Co., Ltd. (Honghu, China). Soy oil was provided by Yihai Kerry (Wuhan) Grain and Oil Industry Co., Ltd. (Wuhan, China). WS (starch content 87.2% and damaged starch content 10.8%) and WG (protein content 81.9%) were purchased from Beijing Ruimai Jiahe Trading Co., Ltd. (Beijing, China). Breadcrumbs (particle size < 2 mm) were purchased from Wuxi Shenglunte International Trade Co., Ltd. (Wuxi, China). SP (protein content 87.4%), EP (protein content 81.6%), and WP (protein content 81.8%) were purchased from Shandong Yuxin Biotechnology Co., Ltd. (Binzhou, China), Dalian Luxue Egg Development Co., Ltd. (Dalian, China), and Henan Shengzhide Trading Co., Ltd. (Zhengzhou, China), respectively. Sudan Red B was purchased from Shanghai Hengdailao Biological Co., Ltd. (Shanghai, China). Sulfuric acid (analytical pure) was purchased from Sinopharm Chemical Reagent Co., Ltd. (Shanghai, China).

### 2.2. Moisture Adsorption Isotherm

WS (1.5 g) and four proteins (WG, SP, EP, and WP; 1.5 g) were individually aliquoted into aluminum cups and then placed in desiccators with a range of controlled water activity (a_w_ at 0.156, 0.343, 0.561, 0.749, and 0.881, established by different concentrations of sulfuric acid solution) at 20 ± 0.5 °C. The aluminum cups were periodically weighed until they reached a constant weight. The moisture content at equilibrium was determined by drying in an oven at 101–105 °C [17]. The relationship between the equilibrium moisture content (EMC) and a_w_ was defined as moisture adsorption isotherm.

### 2.3. Preparation and Characterization of Batters

#### 2.3.1. Batter Preparation

Batters were prepared according to Shan et al. [12] with slight modifications. Three proteins (SP, EP, and WP) at 6% *w*/*w* were individually mixed into the basic batter powder composed of WS (91.7 g) and WG (8.3 g). Deionized water (98 g) was then added to the mixtures by stirring (2000 rpm for 10 min) with an electric mixer (RW20. n, IKA Co., Staufen, Germany) to form homogeneous batters.

#### 2.3.2. Viscosity

Batter viscosity was determined using an NDJ-7 viscometer (Shanghai Jingke Tianmei Trading Co., Ltd., Shanghai, China) according to Sun et al. [18] with slight modifications. The batter was poured into No. II thermostat cup to cover the rotor over 1 mm. The viscometer began to operate after the test temperature reached 25 °C. Data were recorded after the rotor reached the stability. The thermostat cup was washed and dried between uses. Viscosity of batter (η; mPa·s) was calculated according to Equation (1).
η = kA(1)
where k and A represent the coefficient of rotor and the reading of viscometer, respectively.

#### 2.3.3. Rheological Moduli

The rheological behavior of batter was measured using temperature sweep in the oscillation mode of the DHR-2 dynamic rheometer (TA Instruments, New Castle, DE, USA) according to the method given in the published report with slight modifications [6]. An aliquot of 2 g batter was placed on the test platform of rheometer with a cone-plate geometry (60 mm diameter, 2° gap angle). The plates were enclosed within an air escape cover to prevent moisture evaporation. Dynamic rheological behavior of the batters was measured by ramping the temperature from 0 °C to 90 °C at a heating rate of 2 °C/min while applying 0.01 stress amplitude and 1 Hz frequency. The storage modulus (G′), loss modulus (G″), and loss tangent (tan δ = G″/G′) were recorded. The tan δ was used to express the relative viscoelastic characteristics. When the tan δ was less than 1, it represented a dominant elastic feature; if the value was larger than 1, it was characterized by viscosity [19].

#### 2.3.4. Calorimetric Properties

The calorimetric properties of formulated batters were determined using a Q-2000 differential scanning calorimeter (TA Instruments, New Castle, DE, USA). A batter (15 mg) was carefully transferred and hermetically sealed in a stainless-steel pan. An empty pan filled was used as the reference. The pans were equilibrated at room temperature (25 ± 1 °C) for 5 min, and then heated from 20 °C to 130 °C at a rate of 10 °C/min. The onset temperature (T_0_), peak temperature (T_p_), conclusion peak temperature (Tc), and enthalpy change (ΔH) were analyzed using the TA Universal Analysis 2000 software.

### 2.4. BBFN Preparation and Pick-Up

Frozen silver carp surimi was diced into small pieces before being thawed at room temperature (25 ± 1 °C). The thawed surimi (500 g) was finely chopped in a Model HR 7633 chopper mixer (Philips Household Appliances Co., Ltd., Zhuhai, China) with 1200 rpm for 5 min, then 5.0 g NaCl (Food grade) was added, and the chopping was continued for 7 min at 2000 rpm. The chopped surimi was converted into fish nuggets (5 ± 1 g, 2.5 cm × 2.5 cm × 1.5 cm) by cutting with a stainless-steel kitchen knife. The fish nuggets were immediately immersed in the batter for 10 s and then allowed to drain for 15 s. The immersion process was repeated one additional time until liquid drainage was absent, and the fish nuggets were subsequently rolled in breadcrumbs until a uniform coverage was attained.

The pick-up of BBFNs was determined according to the method described by Salvador et al. [14]. The pick-up (W; % *w*/*w*) was defined as the proportion of batter and breadcrumbs coated on fish nuggets, which was calculated according to Equation (2).
(2)$$W\left(\%\right)=\frac{A_{1}-A_{2}}{A_{1}}\times 100$$
where A_1_ and A_2_ represent the total weight of BBFNs (g) and the weight of fish nuggets (g), respectively.

### 2.5. Process and Analyses of Fried BBFNs

#### 2.5.1. Frying Process

BBFNs were fried at 170 °C for 40 s, followed by 190 °C for 30 s using a blast drying oven (101-BS, Shanghai Yuejin Medical Equipment Co., Ltd., Shanghai, China). The thermodynamic properties of crust were then determined after fried BBFNs were drained of excess oil and cooled at room temperature (25 ± 1 °C) for 1 h in a stainless-steel strainer.

#### 2.5.2. Thermogravimetric Analysis (TGA) of Crust

TGA was carried out in a TGA/DSC thermogravimetric analyzer (Mettler-Toledo International Co., Ltd., Shanghai, China). The crust of fried BBFNs was peeled off with a stainless-steel knife and crushed by a QE-50 high speed crusher (Zhejiang Yili Industry and Trade Co., Ltd., Zhejiang, China). The crushed crust (5 g) was weighed into aluminum oxide pans and analyzed by first being heated from 20 °C to 700 °C at a heating rate of 10 °C/min, then cooled from 700 °C to 500 °C at a rate of 20 °C/min under a nitrogen flow rate of 60 mL/min. An empty aluminum oxide pan was used as the reference. The thermal weight (TG) spectrum, recorded using STARe evaluation software, was converted to derivative weight percentage to obtain the DTG curve of crust.

#### 2.5.3. Oil Transport Examined by Optical Microscopy

Dyed oil was prepared by dissolving 1.5 g Sudan red B in 3 L soybean oil and heated at 60 °C for 4 h to obtain a uniform solution [9]. BBFNs supplemented with the added proteins (6%, *w*/*w*) in the batter were fried according to the aforementioned frying conditions. After being cooled to room temperature (25 ± 1 °C), fried BBFNs were cut into thin slices (5 mm × 3 mm × 3 mm) from the junction between the crust and core using a stainless-steel knife. The oil transport phenomenon across the cross section was observed by using an optical microscope (Shanghai BM optical instrument manufacturing Co., Ltd., Shanghai, China) at 4× magnifications in reflective mode.

### 2.6. Statistical Analysis

All samples were tested in triplicate, and the results are expressed as means and standard deviations. Data were processed and analyzed using Origin 8.0 (Origin Lab Corporation, Northampton, MA, USA) and SPSS software (IBM SPSS Version 19, Inc., Armonk, NY, USA). One-way ANOVA and Duncan test were used to determine the variance of sample groups and establish the differences between means at the significance level of *p* < 0.05, respectively.

## 3. Results and Discussion

### 3.1. Moisture Adsorption Isotherms of WS and Proteins

The moisture adsorption isotherm represents the correlation between the moisture content (the weight of water per unit weight of dry matter) and a_w_ at a constant temperature [20], by which the water-binding ability of WS and the four proteins may be reflected. The EMCs of WS and all proteins initially increased slowly at a_w_ < 0.56, then increased rapidly for a_w_ > 0.60 (Figure 1), showing a typical reverse S-shaped curve, consistent with type II isotherm [21,22,23,24,25]. The typical shape of a moisture adsorption isotherm reflects the manner in which the water is bound to the macromolecules. According to Reid et al. [26], water is considered to be “bound water” at a_w_ < 0.70; when a_w_ is greater than 0.70, water is called “free water”. Therefore, the EMCs with a_w_ of 0.15–0.70 may refer to the ability of WS and proteins to bind water. The moisture absorption capacity was the greatest for WS, followed by WG, SP, EP, and WP (Figure 1), which may contribute to the moisture content in fried BBFNs.

### 3.2. Characteristics of Batters

#### 3.2.1. Viscosity

The viscosity of batters is shown in Table 1. WG mainly contains glutenin and gliadin. After absorbing water, the molecular chains of glutenin gradually expand and interweave with each other, which subsequently interact with gliadin to form a gel network, resulting in high batter viscosity [27,28]. Additionally, the damaged starches in WS were expected to be more soluble in water, thereby increasing the batter viscosity [29]. The WS used in the present study had a damaged starch content of 10.8%, favoring the WS swelling power in water. Therefore, the control (without protein added to the batter) had high batter viscosity at 371 mPa·s.

The addition of protein may reduce the content of free water, inhibit the swelling of WS, and weaken the formation of starch gel, leading to a decrease in viscosity [30]. Compared with the control, the batter containing SP had higher viscosity, while the opposite trend was true for EP and WP. Proteins contain hydrophilic groups (such as -COOH, -NH_2_, -OH, and -SH), all of which are capable of forming crosslinks with WS. These crosslinks may be responsible for their higher batter viscosity compared with the batter with only starch [31]. In addition, the high molecular weight proteins had a great propensity to form crosslinks with WS [32]. The molecular weights of SP were a magnitude higher than those of EP and WP [33,34,35]. The high moisture absorption capacity and relative molecular weight of SP may facilitate the cross-linking between SP and WS, leading to an increase in viscosity. Similar results were also observed by Dogan et al. [15].

#### 3.2.2. Rheological Moduli

The rheological behavior of batters is depicted in Figure 2. The G′ and G″ of the control, SP, EP, and WP batters increased slowly at the initial stage of heating (<40 °C) (Figure 2A,B). As the temperature rose, swollen and deformed WS granules would fill in the gel network formed by denatured proteins, promoting cross-linking between protein and starch molecules into a three-dimensional gel with increasing viscoelasticity [36,37]. Compared with the control (around 45 °C), the addition of the three non-cereal proteins delayed starch gelatinization as well as the onset of protein denaturation. Similar results were reported by Zhang et al. [38]. Among the three added proteins, the batter containing SP had the highest G′ and G″, followed by EP and WP, a trend that was consistent with moisture absorption capacity (Figure 1). When the temperature exceeded 65 °C, all the four formulation batters formed a gel, and the viscoelasticity largely remained. The G′ and G″ of the batters containing EP and WP decreased slightly with an increasing temperature, presumably due to the soft “gel” formed by crosslinking of EP/WP and WG [39].

The formation of starch–protein composite gels was the most rapid with the SP treatment, followed by EP, WP, and the control (Figure 2C). When the temperature was less than 32 °C, the tan δ for the control was always above 1.0, and the batter presented a viscous sol. As the temperature was raised, the tan δ began to decrease and eventually fell below 1.0, indicating the critical batter (sol) to gel transformation [40]. For SP-treated batter, the tan δ was less than 1 and a soft “gel” was already formed at the initial stage (<21 °C). As the temperature was increased, the absorption of water by SP will limit the swelling of WS, which facilitated batter gelation [41]. In comparison, the initial gelling temperature for EP- and WP-treated batters was 29 °C and 25 °C, respectively. When the temperature was increased to above 65 °C, the tan δ of all the four batters declined to a distinct and stable low level (<0.25), suggesting that stable starch–protein composite gels were formed [42]. The gel formed by a mixed starch/protein batter could form a protective layer during deep-fat frying, which inhibited moisture evaporation and reduced fat absorption. The oil transport experiment supported this premise (described later).

#### 3.2.3. Calorimetric Properties

The effect of added proteins on batter calorimetric parameters is shown in Table 2. T_0_, T_p_, and T_c_ refer to the temperatures required for the initial, maximum, and conclusion peak values of starch gelatinization or protein denaturation, respectively, and ΔH represents the energy input. In the case of starch, ΔH reflects the breaking of hydrogen bonds, which turns starch from a semi-crystalline state into a soluble state. On the other hand, the ΔH of protein represents the net caloric change involved in the unfolding of the native structure (endothermic process) and the formation of new bonds between protein molecules or intermolecular aggregating (exothermic process) [43]. Compared with the control, the starch gelatinization and protein denaturation temperatures of protein-treated batters showed higher T_0_ and T_p_ and lower ΔH (*p* < 0.05), indicating that the exogenous proteins reinforced thermal stability of the batter. The T_0_ and T_p_ of protein denaturation were the highest for WP treatment, followed by EP and SP, a trend that was nearly opposite with moisture absorption capacity (Figure 1).

The inclusion of these three proteins all reduced the free water of batter and inhibited the gelatinization of starch by limiting the swelling of WS. Denatured polypeptides would form a gel network to envelop some starch granules and inhibit the swelling and gelatinization of starch, leading to an increase in T_0_ and T_p_ and a decrease in ΔH for starch gelatinization [38,44,45,46]. On the other hand, the increased T_0_ and T_p_ for protein denaturation may result from hydrophilic interaction with water, which decreased the free water content and hence resulted in less efficient heat transfer to initiate protein unfolding. The reduced ΔH or energy requirement for protein denaturation by the presence of added proteins promoted batter gelation during deep-fat frying [47]. This result is consistent with the analysis of batter rheological behavior (Figure 2C).

### 3.3. Pick Up of BBFNs

The effect of protein addition on the pick-up of BBFNs is summarized in Table 1. The pick-up for SP batter was the highest, followed by the control, EP, and WP. A positive relationship was observed between the viscosity and pick-up. As the viscosity of batter increased, an increasing amount of batter was adhered to fish nuggets (core), resulting in an increase in pick-up. The high rheological moduli of batter can lead to an increasing pick-up [27] as well as viscosity [12,48]. Compared with the control, the batter containing SP had a higher viscosity and shear moduli (G″, G′), while EP and WP exerted the opposite effect. This explains the higher pick-up for SP batter and lower pick-up for EP and WP batters.

### 3.4. Analyses of Fried BBFNs

#### 3.4.1. TGA of Crust

The TGA of crust is depicted in Figure 3. The TG and DTG curves represent the correlation between weight change and temperature, and between the rate of weight change and temperature, respectively. The peak temperature of the DTG curve represents the temperature of maximum weight loss rate. The process of crust weight loss was divided into two stages: The first stage is the loss of moisture and fat weight; the second stage is the loss of WS and protein gel weight [49]. The thermal stability of crust can be described by the DTG and the TG curves [50]. In the present study, a small peak at roughly 207 °C appeared in the DTG curve of all four crusts, suggesting that the crust started to lose moisture, after which the rate of weight loss increased rapidly. At the initial stage of heating, the temperature corresponding to the maximum rate of weight loss for the three protein-added crusts was higher than that of the control (292 °C). The result may be explained because SP, EP, and WP reinforced the heat stability of crust, thereby delaying moisture evaporation. Among the four types of crust, the moisture loss of the SP crust was the slowest, which is attributed to the high moisture absorption capacity and molecular weight [31,32]. With continuing heating, depolymerization and decomposition of high-molecular weight proteins and WS will occur [51,52], resulting in the loss of WS and protein gel weight. The thermal stability of the crust was significantly affected by the chemical bond energy of the macromolecules between starch and protein [53]. The heat stability of the crust was in the order of SP > WP > EP > control. This was attributed to the increased thermal stability of starch (gelatinization temperature) by the respective proteins (Table 2).

#### 3.4.2. Oil Transport Examined by Optical Microscopy

The migration of frying oil into fried BBFNs was observed by the distribution of oil stained with Sudan Red B dye (Figure 4). The red illumination was confined within the crust rather than the core, and at the interface between the crust and core. These results demonstrated that oil penetration mainly occurred around the crust and did not extend beyond the crust–core interface. The sporadic appearance of red coloring in the interior of core was likely due to oil diffusion through heterogenous, large pores. The greatest extent to which dyed oil penetrated in the crust was observed in the control. For protein-treated batters, the degree of oil penetration, i.e., fat absorption, was in the order of WP > EP > SP, which was consistent with the above batter characteristics analyses, including batter rheology and thermal properties.

## 4. Conclusions

The addition of the proteins significantly affected the viscosity, rheological behavior, and calorimetric properties of a wheat starch-based batter, the pick-up of BBFNs, the thermogravimetric characteristics of crust, and fat absorption (*p* < 0.05). The added proteins acted as water absorbents, inhibited the swelling of WS, and increased the gelatinization temperature due to competitively binding free water in the batter. This phenomenon was most remarkable for the SP batter in which the added protein hindered starch gelatinization. Moreover, the formation of a gel-like structure by the presence of protein additives (SP, EP, WP) in the protein-starch composite batter reinforced the batter thermal stability, resulting in ridged crusts produced upon deep-fat frying. The increased hydrophilic nature, as well as structural hindrances attributed to proteins, was responsible for the decreased fat absorption in BBFNs during deep-fat frying. These results could provide theoretical support for adding protein in breading batters to reduce the fat content of fried BBFNs.

## Figures and Tables

**Figure 1 foods-11-00147-f001:**
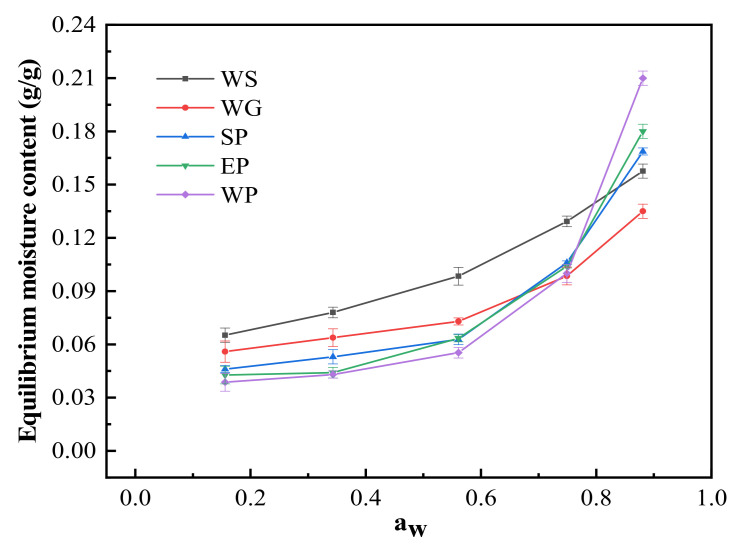
Moisture adsorption isotherms for wheat starch and four proteins. WS, WG, SP, EP, and WP represent wheat starch, wheat gluten, soy protein, egg white protein, and whey protein, respectively.

**Figure 2 foods-11-00147-f002:**
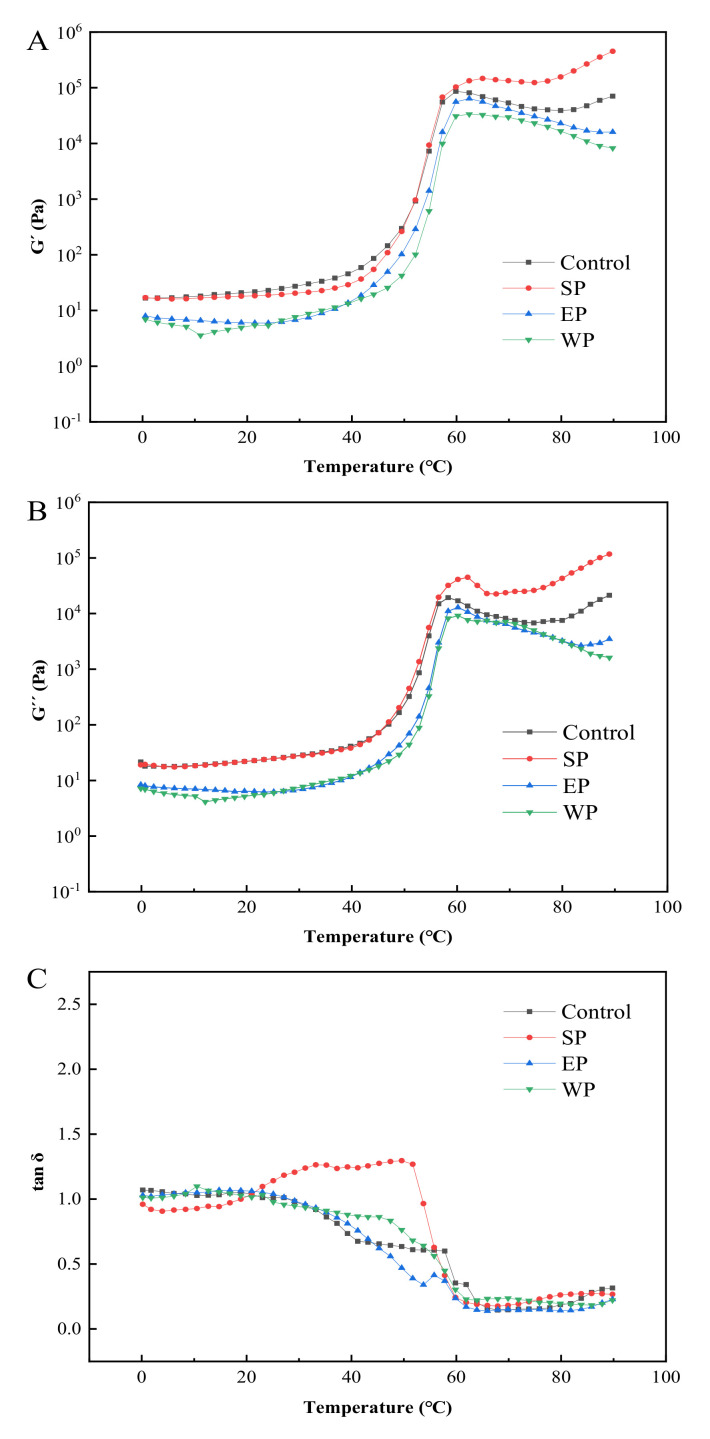
Effect of added proteins on the rheological behavior of batter. (**A**–**C**) refer to the storage modulus (G′), the loss modulus (G″), and the loss tangent (tan δ), respectively; the control refers to the batter without protein added; SP, EP, and WP represent soy protein, egg white protein, and whey protein, respectively.

**Figure 3 foods-11-00147-f003:**
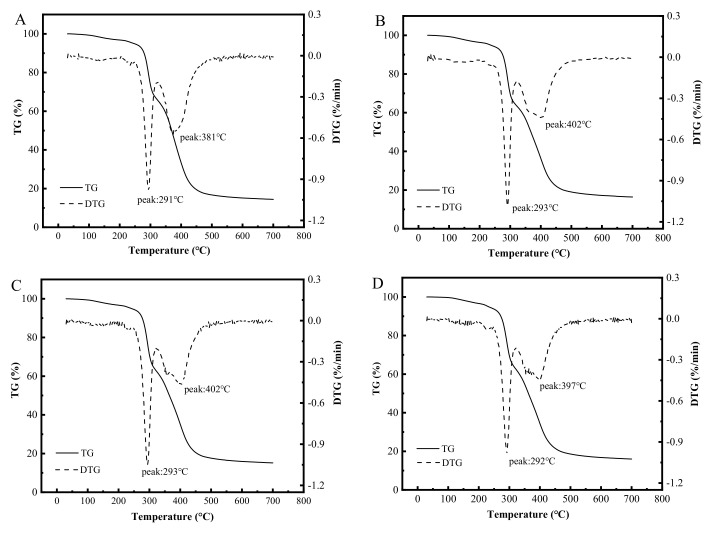
Effect of added proteins on the thermal parameters of crust. (**A**–**D**) represent the control (without protein added to the batter), soy protein, egg white protein, and whey protein, respectively. The TG and DTG curves represent the correlation between weight change and temperature, and between the rate of weight change and temperature, respectively.

**Figure 4 foods-11-00147-f004:**
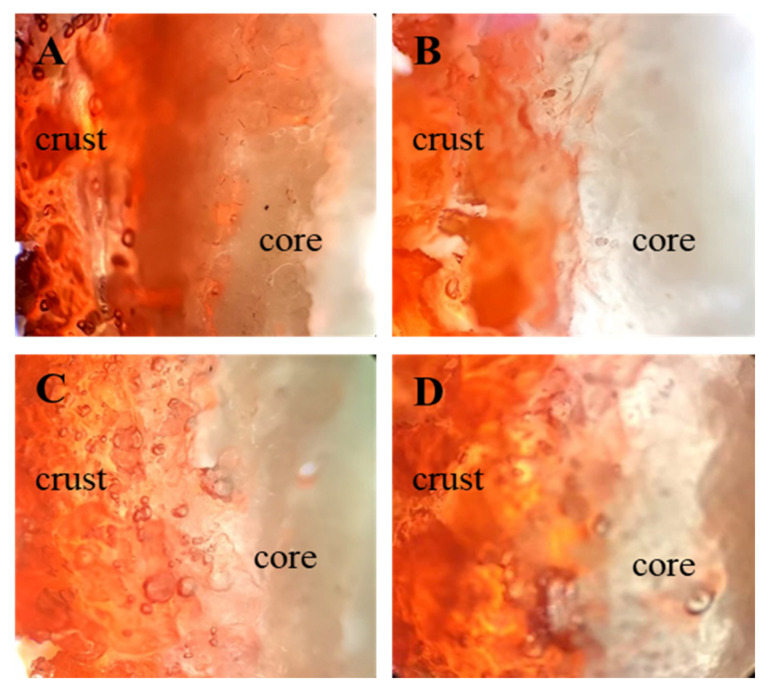
Effect of added proteins on oil penetration of fried BBFNs. All the images were taken at 4× magnification in reflective mode. (**A**–**D**) represent the control (without protein added to the batter), soy protein, egg white protein, and whey protein, respectively.

**Table 1 foods-11-00147-t001:** Effect of the added proteins on the viscosity of batter and the pick-up of BBFNs.

Batter	Viscosity (mPa·s)	Pick-Up (%)
Control	371 ± 6 ^b^	27.9 ± 0.9 ^b^
SP	459 ± 3 ^a^	30.1 ± 0.4 ^a^
EP	327 ± 2 ^d^	25.1 ± 0.7 ^d^
WP	353 ± 4 ^c^	26.1 ± 0.4 ^c^

Data are presented as mean values ± standard deviations from triplicate experiments (*n* = 3). Mean values listed in columns with different letters indicate statistically significant differences (*p* < 0.05).

**Table 2 foods-11-00147-t002:** Effect of added proteins on the calorimetric parameters of batter.

Batter	The Peak of Starch				The Peak of Protein			
	T_0_ (°C)	T_p_ (°C)	T_c_ (°C)	ΔH (J/g)	T_0_ (°C)	T_p_ (°C)	T_c_ (°C)	ΔH (J/g)
Control	52.8 ± 0.5 ^c^	55.9 ± 0.3 ^d^	67 ± 0.1 ^c^	9.8 ± 0.3 ^a^	91.2 ± 0.0 ^c^	95.1 ± 0.2 ^c^	102 ± 0.1 ^c^	1.38 ± 0.0 ^a^
SP	64.4 ± 0.6 ^a^	64.8 ± 0.5 ^a^	76 ± 0.3 ^a^	2.3 ± 0.2 ^b^	104.4 ± 0.0 ^b^	105.8 ± 0.1 ^b^	115 ± 0.2 ^b^	0.4 ± 0.0 ^b^
EP	57.5 ± 0.7 ^b^	62.3 ± 0.3 ^b^	72 ± 0.6 ^b^	1.5 ± 0.1 ^c^	104.4 ± 0.0 ^b^	105.7 ± 0.2 ^b^	114 ± 0.5 ^b^	0.4 ± 0.0 ^b^
WP	56.4 ± 0.4 ^b^	61.4 ± 0.3 ^c^	73 ± 0.2 ^b^	1.6 ± 0.3 ^c^	110.5 ± 0.0 ^a^	110.7 ± 0.0 ^a^	122 ± 0.2 ^a^	0.5 ± 0.0 ^b^

T_0_, T_p_, T_c_, and ΔH represent initial peak temperature, maximum peak temperature, conclusion peak temperature, and enthalpy change, respectively. Data are presented as mean value ± standard deviations from triplicate experiments (*n* = 3). Mean values listed in columns with different letters indicate statistically significant differences (*p* < 0.05).

## Data Availability

The data presented in this study are as described in the individual figures and tables.

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
