# Peer review of "The Effect of Batter Characteristics on Protein-Aided Control of Fat Absorption in Deep-Fried Breaded Fish Nuggets"

_foods, 2022, doi:10.3390/foods11020147_

Round 1
Reviewer 1 Report
The effect of batter characteristics on the protein-aided control of fat absorption in deep-fried breaded fish nuggets was well written.
Few comments suffices
L123 – 124 Specific measurement for the amount of water used should be provided for pans filled with distilled water.
Section 2.3.4. Author(s) should indicate reason for the use of pans filled with distilled water as standard reference pans, rather than empty (aluminium) pans.
Section 3.2.3., L287 - 315. Conclusion temperature (Tc) should be included in this section, discussed and added to Table 2. This is in order to determine the gelatinisation/denaturation temperature range of the starch and protein.
Author Response
Thank you for your comments concerning our manuscript entitled Protein-aided Control of Fat Absorption in Deep-fried Breaded Fish Nuggets: Effect of Batter Characteristics, Foods-1532907.
1. The effect of batter characteristics on the protein-aided control of fat absorption in deep-fried breaded fish nuggets was well written
Response: We have revised the title. Please see the red words highlighted in the revised manuscript.
2. L123 - 124 Specific measurement for the amount of water used should be provided for pans filled with distilled water.
Response: We have revised the manuscript. Please see lines 123-124.
3. Section 2.3.4. Author(s) should indicate reason for the use of pans filled with distilled water as standard reference pans, rather than empty (aluminium) pans.
Response: It is our clericalerror. We have revised the manuscript. Please see lines 123-124.
2. Section 3.2.3., L287 - 315. Conclusion temperature (Tc) should be included in this section, discussed and added to Table 2. This is in order to determine the gelatinisation/denaturation temperature range of the starch and protein.
Response: We have added conclusion temperature (Tc). Please see the Table 2.

Reviewer 2 Report
This is a perfectly solid paper, with a sensible rheological analysis using loss and storage moduli, and gelatinization temperatures. The analysis of protein denaturation is also very satisfactory.
The study of the moisture sorption isotherms is reasonable.
The statistical analysis is also good and reasonable.
Lines 171-172: “mean standard deviation” should be “means and standard deviations”
Author Response
Thank you for your comments concerning our manuscript entitled Protein-aided Control of Fat Absorption in Deep-fried Breaded Fish Nuggets: Effect of Batter Characteristics, Foods-1532907.
1. Lines 171-172: “mean standard deviation” should be “means and standard deviations”
Response: We have revised the sentence. Please see lines 171-172.

Reviewer 3 Report
I have reviewed the manuscript titled: Protein-aided control of fat absorption in deep-fried breaded fish nuggets: Effect of batter characteristics.
This article aims to evaluate the use of soy protein, egg white protein, and whey protein reducing fat absorption in deep-fried breaded fish nuggets comparing to wheat gluten and to analyze the moisture adsorption isotherms, viscosity, rheological behavior, and calorimetric properties of the batters. The information of this work is useful and relevant and the formation of a gel-like structure by the present of these three protein additives in the protein-starch composite batter enhanced the batter thermal stability and formed ridged crusts during deep-fat frying of the manuscript that could be adapted by nugget processing industry especially for fish nuggets in the future. I think the manuscript is acceptable after minor revision. The article is not innovative, however, it contains original and interesting information for surimi processing of fish nuggets. Abstract is well written upon and the wheat starch gelatinization, protein denaturation and crust thermogravimetry temperatures, viscosity and shear moduli of batter are mentioned and evaluated. Introduction is well addressed including fried batter-breaded foods, strategies to decrease fat absorption of fried batter-breaded foods, crust crispness of fried foods, and the effect of wheat starch to wheat protein ratios onmoisture and fat content of deep-fried breaded fish nuggests.
Materials and methods were well described.
This article would be improved if the authors revise the wrong cited reference of Reid et al. (26) at section 3.1, which is showed as Damodaran et al. (2008) in reference section.
I am not a native English speaker. The manuscript seems have no major mistakes are detected and the manuscript can be easily understood. The results are well discussed.
References
All journal references are not following the required format for Foods to use abbreviated journal name. Therefore, it is not acceptable before minor revision as attached file.
I enjoyed reading this manuscript; the needs of special groups of surimi processing of fish nugget and other fried breaded foods high in lipid. This manuscript presents some interesting data.
Date of this review
22 December 2021 9:14

Author Response
Thank you for your comments concerning our manuscript entitled Protein-aided Control of Fat Absorption in Deep-fried Breaded Fish Nuggets: Effect of Batter Characteristics, Foods-1532907.
1. This article would be improved if the authors revise the wrong cited reference of Reid et al. (26) at section 3.1, which is showed as Damodaran et al. (2008) in reference section.
Response: We have revised the reference. Please see line 186.
2. All journal references are not following the required format for Foods to use abbreviated journal name. Therefore, it is not acceptable before minor revision as attached file.
Response: We have revised the references. Please see the red words highlighted in the revised manuscript.
